# StFT: Spatio-temporal Fourier Transformer for Long-term Dynamics Prediction

**Da Long**                                                                    *da.long@utah.edu*
*School of Computing, University of Utah*

**Shandian Zhe**                                                               *u6014464@utah.edu*
*School of Computing, University of Utah*

**Samuel Williams**                                                            *swwilliams@lbl.gov*
*Applied Mathematics and Computational Research Division,*
*Lawrence Berkeley National Laboratory*

**Leonid Oliker**                                                              *loliker@lbl.gov*
*Applied Mathematics and Computational Research Division,*
*Lawrence Berkeley National Laboratory*

**Zhe Bai** *                                                                  *zhebai@lbl.gov*
*Applied Mathematics and Computational Research Division,*
*Lawrence Berkeley National Laboratory*

**Reviewed on OpenReview:** *https://openreview.net/forum?id=o9Cb0ri2oW*

## Abstract

Simulating the long-term dynamics of multi-scale and multi-physics systems poses a significant challenge in understanding complex phenomena across science and engineering. The complexity arises from the intricate interactions between scales and the interplay of diverse physical processes, which manifest in PDEs through coupled, nonlinear terms that govern the evolution of multiple physical fields across scales. Neural operators have shown potential in short-term prediction of such complex spatio-temporal dynamics; however, achieving stable high-fidelity predictions and providing robust uncertainty quantification over extended time horizons remains an open and unsolved area of research. These limitations often lead to stability degradation with rapid error accumulation, particularly in long-term forecasting of systems characterized by multi-scale behaviors involving dynamics of different orders. To address these challenges, we propose an autoregressive Spatio-temporal Fourier Transformer (StFT), in which each transformer block is designed to learn the system dynamics at a distinct scale through a dual-path architecture that integrates frequency-domain and spatio-temporal representations. By leveraging a structured hierarchy of StFT blocks, the resulting model explicitly captures the underlying dynamics across both macro- and micro- spatial scales. Furthermore, a generative residual correction mechanism is introduced to learn a probabilistic refinement temporally while simultaneously quantifying prediction uncertainties, enhancing both the accuracy and reliability of long-term probabilistic forecasting. Evaluations conducted on three benchmark datasets (plasma, fluid, and atmospheric dynamics) demonstrate the advantages of our approach over state-of-the-art ML methods.

---

*Corresponding author (zhebai@lbl.gov)

# 1   Introduction

Predicting the long-term spatio-temporal dynamics of systems governed by partial differential equations (PDEs) is a cornerstone of scientific and engineering research, with broad applications in fields such as earth system modeling, plasma science, fluid dynamics, and beyond. Traditional approaches rely heavily on numerical solvers, which discretize the domain and iteratively solve PDEs using methods including finite element, finite volume and spectral methods (Tadmor, 2012). While effective in many scenarios, these techniques face limitations when applied to multiphysics systems characterized by complex dynamics and multiscale behaviors. They require substantial computational resources and exhibit poor scalability with increasing problem size, rendering them impractical for high-dimensional, large-scale, or long-term physics systems due to excessive computational costs and memory demands.

Recent advances in deep learning have revolutionized the field of PDE modeling by introducing data-driven methodologies that significantly accelerate computations for science while maintaining high accuracy. Inspired by the universal approximation theorem (Chen & Chen, 1995), neural operators that learn the mapping between two function spaces have demonstrated great success in simulating various PDE systems across multiple scientific disciplines without retraining for new conditions (Li et al., 2020; Lu et al., 2021). Building on the success of transformers in natural language processing and computer vision (Vaswani, 2017; Dosovitskiy, 2020), transformer-based neural operators process multiple input functions while enabling arbitrary querying of output function locations, offering enhanced flexibility in handling complex functional mappings (Hao et al., 2023; Li et al., 2022). A series of neural operators have been developed to address complex scientific problems, including weather forecasting, turbulent fluid dynamics, and boiling phenomena (Pathak et al., 2022; Li et al., 2023a; Bi et al., 2023; Hassan et al., 2023; Lin et al., 2021).

Despite the success of these methods, accurate and long-term predictions of complex physical systems remain challenging, primarily due to the requirements for numerical stability, high-fidelity modeling, and reliable uncertainty quantification over extended horizons. The inherent multi-scale nature and multi-physics complexity of such systems necessitate methodologies that can efficiently represent and integrate dynamics across disparate spatial and temporal scales while simultaneously capturing the complex interactions between distinct physical processes, such as the influence of micro-scale turbulence on macro-scale flow in fluids and combustion (Peters, 2009; Natrajan et al., 2007). For large-scale atmospheric pressure systems, high-pressure ridges and low-pressure troughs play a crucial role in shaping local weather patterns; inaccurate representation of those structures can cause significant errors in forecasting rainfall, wind speed, and temperature (Wang et al., 2006; Barlow et al., 2019). In magnetically confined plasmas, multiphysics arises from the coupling of physical processes that govern plasma behavior, including electromagnetic fields, turbulence transport, thermodynamics, and particle interactions that are potentially coupled with kinetic models in high-fidelity simulations. The magnetohydrodynamic (MHD) instabilities caused by current or pressure gradients can limit burning plasma performance, and threaten fusion device integrity (von Goeler et al., 1974; Graves et al., 2012; Seo et al., 2024). Furthermore, integrating uncertainty quantification (UQ) into modeling frameworks is essential for assessing the confidence and reliability of predictions in such complex systems (Cheung et al., 2011; Scher & Messori, 2018; Kruger et al., 2024). Although neural operators present advantages over traditional approaches, they still encounter challenges associated with the demands for scientific fidelity and stability, especially when the underlying physics exhibit rapid changes or high-frequency components. These issues are further intensified in high-resolution simulations of multi-scale scenarios. Recent efforts to address these limitations include $P^2C^2$Net, which encodes a high-order numerical scheme with boundary condition encoding into neural networks (Wang et al., 2024a), and Dyffusion, which trains a forecasting network and an interpolation network that allows for continuous time sampling and multi-step prediction for long-range forecasting (Rühling Cachay et al., 2023). However, most existing neural operators lack built-in mechanisms for uncertainty quantification, which is particularly critical for reliable modeling of long-term dynamics, where even small errors can propagate across scales and result in significant inaccuracies.

Existing approaches for predicting spatio-temporal dynamics can be broadly be classified into two primary categories. The first category comprises models that directly forecast future states at fixed time horizons using a sequence of past observations (Wang et al., 2024b; Kontolati et al., 2024). The second category includes models that utilize an autoregressive manner, which addresses challenges of scaling and fitting complexities

as a continuous-time emulator (Pathak et al., 2022; Rühling Cachay et al., 2023; Lippe et al., 2023; McCabe et al., 2023). Generally, prediction errors incurred in the short term can accumulate, leading to instability and reduced accuracy in long-term forecasts. To mitigate these issues, previous work has proposed techniques such as the pushforward trick, invariance preservation, and iterative refinement (McCabe et al., 2023; Lippe et al., 2023; Brandstetter et al., 2022). Nevertheless, the development of multi-scale modeling frameworks for long-term dynamic prediction remains crucial for capturing the interactions across scales and enhancing prediction accuracy. Concurrently, incorporating uncertainty quantification is critical for identifying the spatial and temporal regions where predictive confidence deteriorates, especially in complex systems where localized uncertainties can influence global dynamics over time.

In this work, we introduce an autoregressive **Spatio-temporal Fourier Transformer (StFT)**, for long-range forecasting of multi-scale and multi-physics systems. At each level of the spatial hierarchy, one StFT block models the physical dynamics associated with a distinct spatial scale or receptive field, as inferred from the spatiotemporal data. Each StFT block adopts a dual-path architecture: (1) **the frequency path** captures large-scale dynamics by operating in the Fourier domain, focusing on low-frequency components that are critical for modeling global behavior. (2) **the spatio-temporal path** operates in the full physical space incorporating all spatio-temporal features to capture fine-scale features. Through a hierarchical composition of StFT blocks across multiple scales, augmented by a generative residual correction block, the resulting model learns the intricate interactions both within the same scales and across different scales. Moreover, it produces uncertainty estimates at each spatial and temporal point, enabling assessment of prediction confidence throughout the forecast. The cascading StFT blocks enable our model to predict high-resolution dynamics across a spectrum of varying scales in correlated physical processes. By integrating StFT within an auto-regressive framework, our method achieves superior accuracy in long-term predictions compared to existing state-of-the-art autoregressive baselines. Our contributions are summarized as follows:

- We propose Spatio-temporal Fourier transformer (StFT), a novel ML model that learns underlying dynamics across spatial scales for multi-physics systems via a dual-path (frequency and spatio-temporal path) architecture, which effectively captures both the global, large-scale structures and local, fine-scale features.

- We propose StFT-F, which incorporates a probabilistic residual correction mechanism to refine the forecasting of StFT temporally and provide pointwise uncertainty quantification.

- We propose an overlapping tokenizer and a detokenizer that share regions between adjacent patches, improving spatial smoothing and reducing discontinuity artifacts.

- We demonstrate the effectiveness of StFT in an autoregressive framework on a diverse set of applications including the plasma, fluid, and atmospheric dynamics. Evaluating performance across variables, StFT outperforms the best baselines. Its probabilistic variant, StFT-F further improves average forecasting accuracy by 5% and produces uncertainty estimates that are empirically calibrated, as demonstrated through confidence-based evaluation.

## 2 Related Work

**Neural Operators.** Neural operator architectures and their variants have been proposed, including Fourier neural operators (Li et al., 2020; Gupta et al., 2021; Tran et al., 2021; Cao et al., 2023; Li et al., 2023b; Rahman et al., 2022), DeepONet (Lu et al., 2021; Wang et al., 2021; Jin et al., 2022; Wang et al., 2022; Kontolati et al., 2024; Prasthofer et al., 2022), transformer based operators (Hao et al., 2023; Cao, 2021; Li et al., 2022), and image-to-image operators (Gupta & Brandstetter, 2022; Long et al., 2024). U-Net, a fundamentally hierarchical structure model, which has inspired several neural operators (Rahman et al., 2022; Liu et al., 2022b; Gupta & Brandstetter, 2022), allows solutions to multi-scale PDEs by hierarchically aggregating feature representations of progressively coarser spatial resolutions. Recent work in computer vision (Liu et al., 2021; Fan et al., 2021; Zhang et al., 2022) have introduced methods for extracting multi-scale features through hierarchical architectures. However, these hierarchical models do not explicitly model or forecast the multi-scale structures of physical processes, which limit the transparency and interpretability of

their representations across scales. In contrast, our method begins with a coarse approximation that captures large-scale, low-frequency phenomena, and incrementally refines the representation over layers to resolve finer details. These structured decompositions enable error diagnosis, enhance interpretability of model performance of different scales, and allow for targeted improvements with an explicit refinement mechanism.

**Generative Models.** Generative models, especially diffusion models have demonstrated success in various domains, including vision, audio, robotics (Ho et al., 2020; Song et al., 2020; Tian et al., 2024; Kong et al., 2020; Wolleb et al., 2022), and relevance to spatio-temporal dynamics prediction (Ho et al., 2022; Voleti et al., 2022; Singer et al., 2022; Rühling Cachay et al., 2023). As an alternative approach in generative modeling, flow matching has been introduced to support efficient sampling and has since been applied to video generation (Lipman et al., 2022; Liu et al., 2022a; Albergo & Vanden-Eijnden, 2022; Polyak et al., 2024; Esser et al., 2024). While video generation typically explores a range of creative and diverse possibilities from text or image prompts, forecasting spatial-temporal dynamics driven by PDEs necessitates more than mere statistical resemblance - it requires each prediction is firmly grounded in the underlying physics. To achieve accurate forecasting while capturing the inherent stochasticity of physical processes, our work incorporates a flow matching block following the proceeding StFT blocks. This enables the model to align its prediction distribution with the underlying physical dynamics and generate calibrated uncertainty estimates via confidence-based metrics.

## 3    Method

**Formulation.**    We consider an autoregressive formulation for long-term multi-scale spatiotemporal physical processes. We define a vector $\tilde{u}_t$ representing the historical snapshots of the multi-physics variables at timestamps from $t - k + 1$ to $t$, in a total of $k$ snapshots of $\tilde{u}_t = [u_t, u_{t-1}, \ldots, u_{t-k+1}]$ specifically. We formulate the probabilistic one-step forward neural operator StFT-F as

$$u_{t+1} = \mathcal{F}_{\theta_d}(\tilde{u}_t) + r_{t+1}, r_{t+1} \sim \mathcal{P}_{\theta_g}(r|\tilde{u}_t, \mathcal{F}_{\theta_d}(\tilde{u}_t)), \tag{1}$$

where $\mathcal{F}_{\theta_d}$ denote the StFT operator, a deterministic forecasting parameterized by $\theta_d$, and $\mathcal{P}_{\theta_g}$ is the generative flow matching block parameterized by $\theta_g$ for refining the forecasting of StFT while quantifying uncertainty estimates. $\mathcal{F}_{\theta_d}$ represents the deterministic evolution of the system that encapsulates the dynamics from multi-scale spatial refinement in StFT. The residual refinement $\mathcal{P}_{\theta_g}(r|\tilde{u}_t, \mathcal{F}_{\theta_d}(\tilde{u}_t))$ captures the probabilistic nature of the residual from the generative model. It represents a prediction distribution conditioned on the current state $\tilde{u}_t$ and the deterministic prediction $\mathcal{F}_{\theta_d}(\tilde{u}_t)$, modeling the uncertainty or variations that missed by the deterministic component. The residual $r_{t+1}$ calculates deviations from the deterministic prediction, and its distribution allows the model to account for noise or inherent stochasticity in the physical processes. Therefore, by sampling residual $r_t$, our model learns stochastic trajectories from data. Besides providing prediction uncertainties, these stochastic trajectories can help study the long-term behavior, stabilities, and bifurcations in stochastic systems  (Lucor et al., 2003; Krämer et al., 2022).

Figure 1 presents the overview of the StFT-F, the overlapping tokenizer, and the design of the StFT block. Algorithm  1 and  2 detail the design of our model. The following subsections introduce StFT with the overlapping tokenizer/detokenizer and the residual correction mechanism based on flow matching.

### 3.1   StFT Block

**Overlapping tokenizer.** Each StFT block first tokenizes the discretized functions $\tilde{u}_t$ of shape $T \times W \times H \times C$, where $T$ is the temporal dimension, $W, H$ are the spatial dimensions, and $C$ denotes the number of physical variables. We apply tokenization along the spatial dimensions for each variable. To enhance spatial continuity while minimizing visual artifacts, we propose to use an overlapping tokenizer (OLT) and detokenizer (OLDT) that allows adjacent patches to share boundaries through overlapping regions. For instance, as shown in Figure 1, a $3 \times 3$ input generates four $2 \times 2$ patches with a $1 \times 1$ overlap, where the overlapping areas (indicated in gray) are shared between patches. During detokenization, overlapping regions are reconstructed by averaging the corresponding values from neighboring patches. This strategy effectively mitigates discontinuity issues at the patch boundaries, which is particularly important for accurately representing smooth and

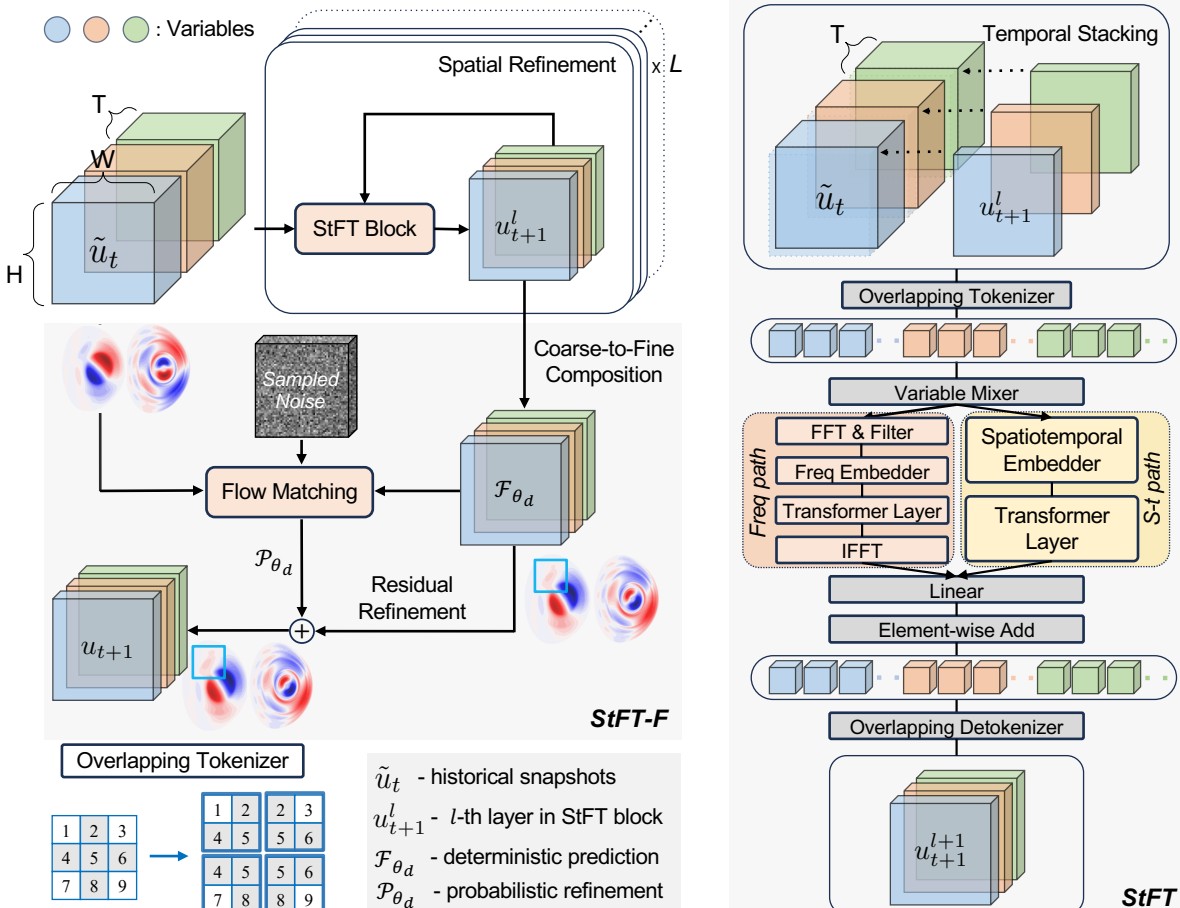

Figure 1: Left: **Overview of the proposed StFT and StFT-F model**. The model predicts $u_{t+1}$ using the past $k$ snapshots $\tilde{u}_t$, employing $L$ spatial refinements from coarse to fine scales through the proposed Spatiotemporal Fourier Transformer (StFT) blocks. Bottom left: an illustration of the overlapping tokenizer, where the patch size is $2 \times 2$, and the overlapping number is $1 \times 1$. Right: **Illustration of the proposed StFT block**. First, the past snapshots $\tilde{u}_t$ and a coarser prediction from the previous layer are temporally stacked. The stacked discretizations are passed to the overlapping tokenizer to generate tokens for each variable. Next, tokens corresponding to different variables in the same spatial are mixed through a variable mixer. Two paths of transformation, frequency path and spatiotemporal path, process the frequency embeddings and the spatiotemporal embeddings respectively. The finer prediction for timestamp $t + 1$ is obtained after passing through the overlapping detokenizer.

continuous target functions. Furthermore, incorporating shared boundaries into patch embeddings enriches feature representation and extraction from fine-scale structures.

**Variable mixer.** Each StFT block is designed to handle a specific scale; therefore, by employing a specific patch size, we partition the input at a corresponding level of granularity. To ensure that the first block captures the coarsest features or the largest scale, we set the patch size $p_w^1 \times p_h^1$ to a large value, allowing it to model broad spatial structures effectively. As a result, $\mathcal{O}(\frac{W}{p_w^1} \times \frac{H}{P_h^1} \times C)$ patches are fed into the variable mixer, where patches corresponding to different physical processes but sharing the same spatial domain are mixed into a single token. Following this step, two transformation paths are performed: one is in the spatio-temporal domain, which operates self-attention on spatio-temporal embeddings; the other is in the frequency domain, which operates on frequency embeddings.

**Frequency embeddings.** The tokens are first processed by a 2D/3D Fourier transform, where only low-frequency components are retained. These low-frequency components are then passed through a frequency embedder to obtain frequency embeddings $f_t$. Subsequently, these frequency embeddings are fed to the

standard transformer layers for mixing information and nonlinear transformation in the frequency domain. Finally, an inverse 2D/3D Fourier transform and a linear projection are applied to map the frequency embeddings back to the spatio-temporal domain.

**Spatio-temporal embeddings.** The same set of tokens first pass through a spatio-temporal embedder, after which the spatio-temporal embeddings $e_t$ are processed by multiple standard transformer layers for mixing correlations and nonlinear transformation. Finally a separate linear projection is applied to get predictions for each patch of $u_{t+1}$.

Next, an overlapping detokenizer yields the first block prediction $u_{t+1}^1$. Each token represents a significant portion of the historical snapshots, encapsulating macroscopic structural features. This coarse-level partitioning reduces the complexity of modeling fine-grained details. By maintaining a lower granularity, the model prioritizes structural coherence over extraneous details, enabling it to focus on capturing and predicting global relationships between regions more effectively.

## 3.2   A Hierarchy of StFT Blocks

In the subsequent StFT blocks, we shift our focus to smaller scales with fine details. Consequently, we concatenate the prediction $u_{t+1}^1$ with the input $\tilde{u}_t$, and consider this combination as the input for the next StFT block. We further subdivide each patch from the previous StFT block into smaller patches. The smaller patches allow for less information to be aggregated within a single patch, thereby minimizing the risk of losing local variations and enhancing the richness and informativeness of the fine-scale representation. By leveraging the finer granularity of the patches to focus on smaller regions, it allows the model to better localize features and capture their details. In addition, conditioning on the coarser prediction allows the models to iteratively refine its estimates, beginning with a broad global summary of the corresponding regions. Through the repeated subdivision of the patches, the model progressively refines its predictions across multiple scales. As shown in Figure 1, the StFT model predicts $u_{t+1}$ and applies a total of $L$ multi-scale spatial refinements.

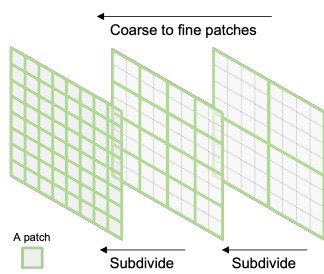

Figure 2: A patch is subdivided into smaller patches for a hierarchical learning.

## 3.3   Residual Refinement and Uncertainty Estimation Based on Flow Matching

Finally, the model refines its deterministic predictions through a rectified flow block, which belongs to the family of flow matching models (Liu, 2022; Liu et al., 2022a). Flow matching is formulated as an ordinary differential equation in time $\tau \in [0, 1]$, $\frac{d}{d\tau}\psi_\tau(x) = \nu_\tau(\psi_\tau(x))$, where the learnable velocity field $\nu_\tau$ directs the transformation of each sample $X_0$ from a source distribution $p_0$, typically a Gaussian distribution, toward the target distribution $X_1 \sim p_1$ with $p_1$ representing the data distribution. If we prescribe the velocity field $\nu_\tau$ such that it guides every sample along a straight-line trajectory from $X_0$ to $X_1$, it is referred to as a rectified flow. In this case, $X_\tau$ represents the linear interpolation across the entire timespan between $X_0$ and $X_1$, which can be expressed as $X_\tau = \tau X_1 + (1 - \tau)X_0$. We employ a parameterized $\mathcal{M}_{\theta_g}$ to approximate $\nu_\tau$, leading to the following learning objective: $\mathcal{L}(\theta_g) = \mathbb{E}_{\tau, X_0, X_1} \left\| \mathcal{M}_{\theta_g}(X_\tau, \tau) - (X_1 - X_0) \right\|^2$. In our model, the rectified flow block takes the deterministic prediction $u_{t+1}$ from the composition of $L$ StFT blocks and the observations $\tilde{u}_t$ as conditioning inputs. Its objective is to generate the distribution of residuals $r_{t+1} = y - \sum_j u_{t+1}^j$, where $y$ is the ground truth for the solution at $t + 1$. Our training loss then becomes:

$$\mathbb{E}_{X_0 \sim \mathbf{N}(0,\mathbf{I}), \tau \sim (0,1)}[(\mathcal{M}_{\theta_g}(\tilde{u}_t, \mathcal{F}_{\theta_d}(\tilde{u}_t), \tau, X_\tau) - (r_{t+1} - X_0))^2], \tag{2}$$

where $X_\tau$ is the linear interpolation between the source sample $X_0$ and the target $r_{t+1}$.

**Algorithm 1: StFT**

**1 Inputs:** history $\tilde{u}_t = (u_t, \ldots, u_{t-k+1})$
**2 Initialize:** blocks $l \in [1, L]$, patch sizes $p_{h_l, w_l}$, truncation modes $m_{h_l, w_l}$, overlaps $o_{h_l, w_l}$, $u_{t+1}^0$ as None, $u_{t+1}$ as **0**
**3** $v \leftarrow$ var idx
**4 for** $l = 1, \ldots, L$ **do**
**5**    $x_t \leftarrow \text{TemporalStacking}(\tilde{u}_t, u_{t+1}^{l-1})$;
**6**    $\{x_t\}_v \leftarrow \text{OLT}(x_t, p_{h,w}, o_{h,w})$;
**7**    $\{x_{t+1}\}_v$ : **Invoke *Freq.* & *S-t.* paths**
**8**    $u_{t+1}^l \leftarrow \text{OLDT}(\{x_{t+1}\}_v, p_{h,w}, o_{h,w})$
**9**    $u_{t+1} \leftarrow u_{t+1} + u_{t+1}^l$
**10 Return:** $u_{t+1}$

**Algorithm 2: StFT block: *Freq.* and *S-t.* paths**

**1 Frequency Path:** $\bar{x}_{t,1} \leftarrow \text{VariableMixer}^1(\{x_t\}_v)$
**2** $f_t \leftarrow \text{FFTFilter}(\bar{x}_{t,1}, m_h, m_w)$
**3** $f_{i,t} \leftarrow \text{FreqEmbedder}(f_{i,t})$
**4** $f_t \leftarrow \text{TransformerBlock}^1(f_t)$
**5** $\{x_{t+1,1}\}_v \leftarrow \text{iFFT}(f_t, m_h, m_w)$
**6** $\{x_{t+1,1}\}_v \leftarrow \text{Linear}^1(\{x_{t+1,1}\}_v)$
**7 Spatiotemporal Path:** $\bar{x}_{t,2} \leftarrow \text{VariableMixer}^2(\{x_t\}_v)$
**8** $e_t \leftarrow \text{StEmbedder}(\bar{x}_{t,2})$
**9** $e_t \leftarrow \text{TransformerBlock}^2(e_t)$
**10** $\{x_{t+1,2}\}_v \leftarrow \text{Linear}^2(e_t)$
**11 Merge:** $\{x_{t+1}\}_v \leftarrow \{x_{t+1,1}\}_v + \{x_{t+1,2}\}_v$

## 4 Experiments

### 4.1 Datasets

In this section, we consider three spatio-temporal multi-physics systems arising from time-dependent PDEs of a variety of complexities, including a high-dimensional plasma dynamics system based on reconstructed equilibrium of DIII-D experimental discharges (Bai et al., 2025), a 2D incompressible Navier-Stokes equation in velocity-pressure form within a square domain driven by an external force, and a viscous shallow-water equation modeling the dynamics of large-scale atmospheric flows on a spherical domain. The problem setup and data generation are detailed in Appendix B.

### 4.2 Experimental Setup

**Long-term multi-physics prediction up to a horizon of 244 timesteps.** Our goal is to simulate long-time trajectories given a few initial observations. This task is particularly challenging due to the multiple correlated variables present in the Navier-Stokes and plasma magnetohydrodynamics (MHD), with the test trajectories consisting of snapshots that vary from 71 to 244. We employ an autoregressive framework for all the methods: during training, each model utilizes five historical snapshots to predict the next one in a forward pass. At test time, given the initial five snapshots of a trajectory, all models autoregressively generate the entire trajectories by iteratively predicting future states based on their own previous outputs.

**Baseline setup.** We evaluate these datasets using the following well-known and state-of-the-art methods for comparison: autoregressive AFNO, autoregressive Fourier Neural Operator (AR-FNO), autoregressive U-Net (AR-UNet), and autoregressive vision transformer (AR-ViT). More specifically, for AFNO, we used the latest variant from the DPOT work (Hao et al., 2024), which is enhanced with a temporal aggregation layer and improved expressivity through the removal of enforced sparsity. For FNO, we use the authors' open-source implementation. For U-Net, we employ the implementation of the modified U-Net as evaluated in the recent BubbleML work (Hassan et al., 2023), where the modified U-Net was initially used in PDEBench work (Takamoto et al., 2022), and demonstrated superior performance over their baselines.

**Validation and hyperparameter tuning.** We divide the trajectories of each dataset into training, test, and validation sets. For each method, we identify the tunable hyperparameters, specify a range for each hyperparameter, and conduct a grid search. For our methods, we implement both StFT, the deterministic component of our model, and StFT-F, our model with a generative residual refinement block. We use the AdamW optimizer with a learning rate of $1e^{-4}$ to train those models on an A100 GPU. We ensure that all models are fairly and thoroughly trained by imposing an identical time budget across all models. More specifically, we impose 24 GPU hours on the plasma MHD and Navier-Stokes datasets, and 48 GPU hours on the shallow-water dataset. A comprehensive list of all the hyperparameters along with their respective ranges is provided in the Appendix F.

**Evaluation metrics.** First, we evaluate the forecasting performance by calculating the mean $L_2$ relative error. For StFT-F, in order to obtain the mean prediction, we generate 50 stochastic predictions at each

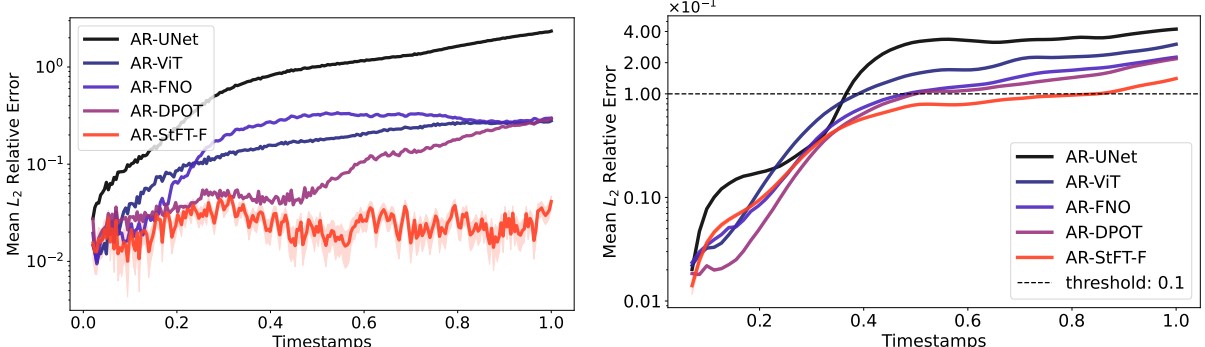

Figure 3: Results of autoregressive prediction in $L_2$ relative error (log scale) across the timespan: (left) perturbed parallel vector potential $\delta A_\parallel$ in plasma MHD, and (right) magnitude of velocity in shallow-water equation. The shaded region indicates the uncertainty distribution of $\sigma$ in the relative error of StFT-F. For a given error threshold, StFT-F maintains accuracy over at least twice the time horizon compared to the baselines.

Table 1: Quantitative results of our model and baselines in the same autoregressive framework: relative $L_2$ error over three spatiotemporal prediction systems. AR-StFT refers to the deterministic model's results. AR-StFT-F denotes the probabilistic model with residual refinement of StFT. All models have been subjected to hyperparameter tuning to ensure fair and optimal performance comparisons.

| Dataset | Variable(s) | AR-StFT | AR-StFT-F | AR-DPOT | AR-FNO | AR-ViT | AR-UNet$_b$ |
|---------|-------------|---------|-----------|---------|--------|--------|-------------|
| Plasma MHD | $\delta\phi$ | 2.80e-2 | 2.24e-2 | 1.04e-1 | 2.28e-1 | 1.73e-1 | 1.02e0 |
| | $\delta A_\parallel$ | 2.45e-2 | 2.30e-2 | 8.36e-2 | 2.30e-1 | 3.24e-1 | 8.13e-1 |
| | $\delta B_\parallel$ | 3.05e-2 | 2.66e-2 | 8.98e-2 | 2.33e-1 | 1.95e-1 | 7.79e-1 |
| | $\delta n_e$ | 2.84e-2 | 2.45e-2 | 8.64e-2 | 2.33e-1 | 2.08e-1 | 1.01e0 |
| | $\delta n_i$ | 3.28e-2 | 2.93e-2 | 8.76e-2 | 2.33e-1 | 2.18e-1 | 1.04e0 |
| | $\delta u_e$ | 3.99e-2 | 3.73e-2 | 9.59e-2 | 3.18e-1 | 2.99e-1 | 6.96e-1 |
| Navier-Stokes | $u$ | 3.38e-2 | 3.30e-2 | 4.67e-2 | 4.46e-2 | 5.09e-2 | 6.16e-2 |
| | $v$ | 3.60e-2 | 3.17e-2 | 4.52e-2 | 4.57e-2 | 4.60e-2 | 6.15e-2 |
| | $p$ | 5.16e-2 | 4.44e-2 | 6.18e-2 | 5.90e-2 | 7.03e-2 | 7.84e-2 |
| Shallow-Water | **V** | 6.25e-2 | 6.53e-2 | 7.97e-2 | 9.53e-2 | 1.33e-1 | 2.02e-1 |

autoregressive step, and then inject their mean into the next step. Additionally, we assess the uncertainty quantification capability of StFT-F by sampling 100 trajectories for each test case. For each trajectory, at each autoregressive step, we generate a single prediction, which is subsequently fed into the next autoregressive step to iteratively forecast the full sequence.

## 4.3 Main Results

Table 1 shows the forecasting performance of all the models on the three applications. We present several test trajectories and visualize their uncertainties across the forecasting time horizon estimated by StFT-F, as illustrated in Figure 5. StFT performs significantly better than all other baselines across all physical processes in the three applications. In the plasma MHD dataset, the test trajectory has a total of six coupled physics variables and 244 snapshots. On average, StFT achieves a reduction in error by a factor of three compared to the best baseline, AR-DPOT(AFNO). Although AR-FNO maintains a high resolution in its long-term prediction, it fails to capture the correct dynamics of mode evolution, leading to out-of-phase predictions as shown in Figure 4.

We examine the error growth by plotting the $L_2$ relative errors over time, as illustrated in Figure 3 for several representative variables. In the bottom figure for the shallow-water dataset, AR-FNO first appears to slightly better than all other methods during the short term from timestamp 0 to 0.2, and StFT-F shows superior performance soon after. For the plasma dataset, StFT-F demonstrates dominance starting from $t = 0.2$ with

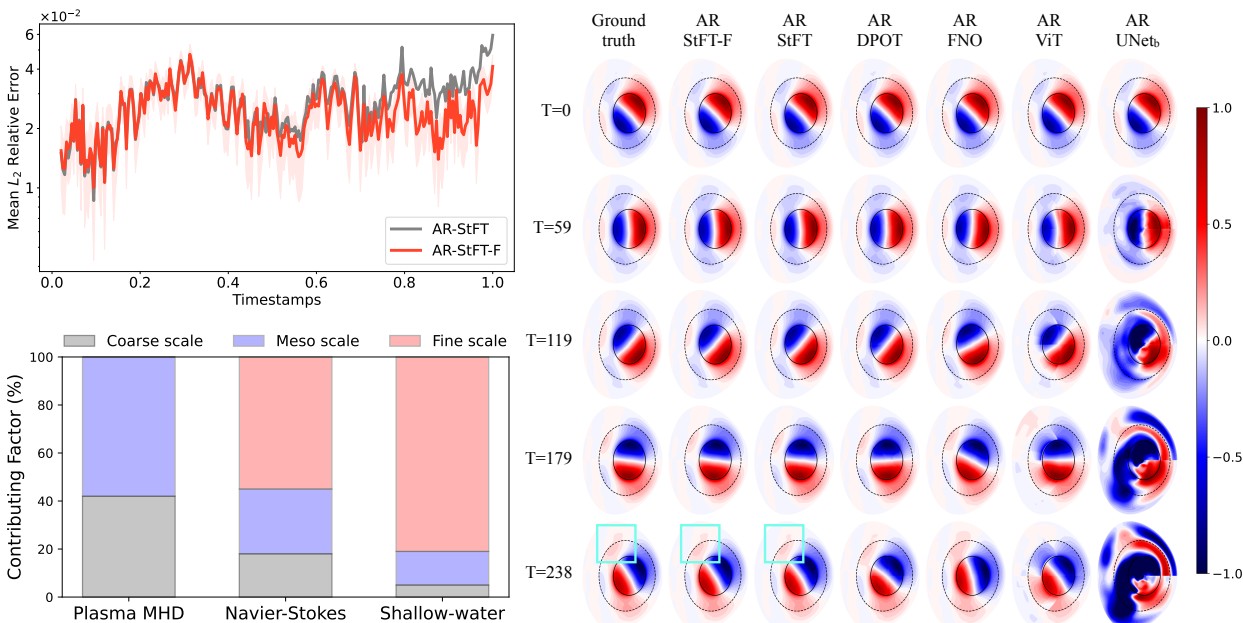

Figure 4: Left top: Comparing StFT and StFT-F in $L_2$ relative error across timestamps for two plasma MHD variables with uncertainty bands of StFT-F. Left bottom: The contribution of each StFT block. From bottom to top, the patch size decreases. For this plasma data, the first two levels are sufficient to capture the multi-scale structures, whereas for Navier-Stokes and shallow-water equations, the finest scale contributes more significantly. Right: Spatiotemporal evolution of perturbed electrostatic potential $\delta\phi$ predicted by autoregressive models. StFT and StFT-F remain accurate past $T = 59$, whereas baseline methods exhibit out-of-phase predictions.

a stable performance, while the errors of all baseline methods begin to propagate from that point onward, resulting in a rapid decline compared to StFT-F. Notably, StFT-F exhibits long-term stability relative to the other methods. As shown in Figure 4, both of our methods accurately capture the dominant mode phase, and StFT-F generates predictions that are more closely aligned with the ground truth compared to StFT. We also compare the error over time comparing StFT and StFT-F. StFT-F begins to prevail from $t = 0.6$. For the shallow-water and Navier-Stokes datasets, on average, StFT reduces the errors by an average of 27% and 25%, respectively. In Navier-Stokes and plasma datasets, StFT-F not only surpasses StFT but also offers the additional capability of uncertainty quantification, achieving error reductions of 10% in both cases. In the shallow-water dataset, we observe a slight increase in error with StFT-F. These results indicate that our method achieves superior long-term stability and accuracy among all other methods. More visualizations and results are included in Appendix G.

### 4.4 Uncertainty Quantification

Figure 5 presents the distribution of the empirical standard deviation along with the mean prediction. As observed, regions with large errors correspond to those exhibiting significant uncertainties predicted by StFT-F. Additionally, it is evident that uncertainties increase with time. This aligns with our expectation, as errors accumulate during the autoregressive forecasting process. Besides the empirical evaluation relying on the standard deviation plots, we further measure the ro-

Table 2: Average confidence interval (90% and 95%) coverage.

| Dataset | CI: 90% | CI: 95% |
|---|---|---|
| Plasma MHD | 89.4% | 92.5% |
| Shallow-Water | 89.5% | 98.0% |

bustness of StFT's uncertainty quantification using confidence intervals as shown in Table 2. Specifically, we use the predicted uncertainty to compute empirical coverage by measuring the proportion of ground truth values that fall within a certain confidence interval around the predicted mean. This provides a more rigorous evaluation of StFT's ability to capture uncertainties. Details about CI coverage and results regarding each

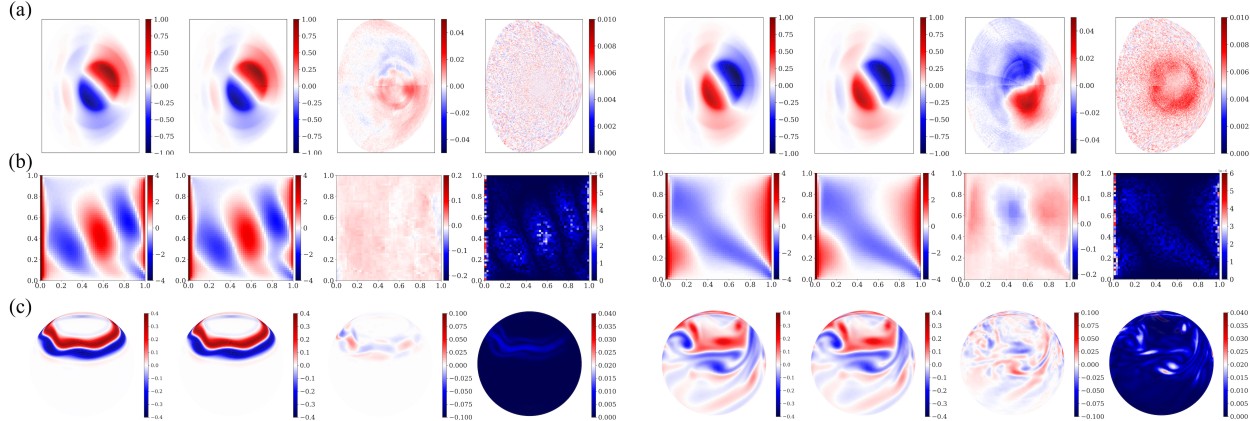

Figure 5: Evaluation of forecasting for three applications: ground truth, StFT-F prediction, residual, and uncertainty over time — shown for initial (left) and final (right) states. Variables include: (a) $\delta\phi$ in plasma MHD, (b) $u$ in Navier-Stokes, and (c) $\mathbf{V}$ in shallow-water equations.

variable are included in Appendix E. For the 90% confidence interval, average coverage is very close to ideal value of 90%, demonstrating that StFT-F is well-calibrated around the 90% confidence interval.

## 4.5 Ablation Study

**Multi-scale and frequency path.** To evaluate the effectiveness of the hierarchical structure and the frequency path in StFT, we conduct an ablation study employing mono-scale versus multi-scale models, and employing models with and without the frequency path. Experimental results are shown in Table 3a.

**Convergence study on number of scales.** We evaluate the performance of StFT on the number of scales for the plasma MHD dataset. As shown in Table 3b, it is observed that increasing the number of scales leads to a reduction in prediction error to a certain point, with the two-layer configuration achieving the lowest error. The inclusion of additional scales results in a slight error increase, indicating a conver-

Table 3: Ablation study of StFT. (a) Evaluates model structure and frequency path. (b) Shows performance impact of using more hierarchical scales.

(a) Effect of mono-/multi-scale structures and frequency path $\mathcal{F}$.

| Model | Plasma MHD | Shallow-Water |
|---|---|---|
| Mono-scale + $\mathcal{F}$ | 0.0805 / 0.105 | 2.5729 / 0.101 / 0.0975 |
| Multi-scale | 0.0404 | 0.0956 |
| Multi-scale + $\mathcal{F}$ | **0.0307** | **0.0625** |

(b) L2 error vs. number of scales (plasma MHD).

| # Scales | L2 Rel. Error |
|---|---|
| 1 | 0.0805 |
| 2 | **0.0307** |
| 3 | 0.0385 |
| 4 | 0.0391 |

gence in model performance. Additionally, we conduct an ablation study on the overlapping tokenizer. More details are included in Appendix C.

## 4.6 Additional Results

**Computational and Space Complexity.** We compare StFT with other baselines regarding the inference FLOPS per sample, training time and inference time per batch, and peak memory usage during training. We also compare StFT and StFT without the overlapping tokenizer, and we observed this enhancement comes with negligible impact on compute cost, as in Appendix D.

**Contribution of Each Scale.** To assess the contribution of each StFT block for a specific scale in fitting the training data, we quantify the weight of each block by: $\mathcal{W}_i = \frac{\|\mathbf{y}_i\|_2}{\|\mathbf{y}\|_2}$, where $\mathbf{y}_i$ represents the prediction from the $i$-th StFT block, and $\mathbf{y}$ denotes the ground truth. We normalize the contributions, and present the

contributions in Figure 4. A greater contributing factor from the fine-scale layer in StFT is observed in the Navier-Stokes and shallow-water equations, attributed to the sharper changes and smaller scale structures inherent in the dynamics of higher-order nonlinearities.

## 5 Conclusion

In this paper, we propose a spatio-temporal Fourier transformer (StFT) for multi-scale and multi-physics long-term dynamics forecasting. Specifically, each StFT block is tailored to address a particular spatial scale, and through a hierarchical composition of multiple StFT blocks spanning different scales, StFT learns the interplay between multiple scales and interactions between multiple physical processes, resulting in stable and accurate long-term dynamics forecasting in an autoregressive manner. Furthermore, we propose and demonstrate the use of a generative residual correction mechanism, which enables meaningful quantification of uncertainties in the predictive model. Despite demonstrating superior forecasting ability in SciML, the model is based on regular grids, which constraints its applicability to irregular grids, and future work will focus on extending the framework to handle irregular geometries. The potential in improving its performance includes model parallelism across the multi-scale StFT blocks and extending StFT-F to end-to-end training.

## Acknowledgements

ZB, DL, SW, and LO acknowledge support from the U.S. Department of Energy, Office of Science, Sci-DAC/Advanced Scientific Computing Research under Award Number DE-AC02-05CH11231 with DL receiving additional support from the Margolis Foundation and MURI AFOSR under grant FA9550-20-1-035. SZ was supported by MURI AFOSR grant FA9550-20-1-035, Margolis Foundation, and NSF Career Award IIS-2046295. This research used resources of the National Energy Research Scientific Computing Center (NERSC), a U.S. Department of Energy Office of Science User Facility located at Lawrence Berkeley National Laboratory, operated under Contract Number DE-AC02-05CH11231, as well as resources from the NCSA Delta GPU cluster from NCF Access program under award number IIS-2046295.

## Broader Impact Statement

Turbulence remains one of the great unsolved problems in physics. Yet turbulence, whether driven by gravity, heating, or magnetic fields, manifests in physical phenomena spanning multi-scale fluid flow to plasma fusion reactors to planetary atmospheres to convective layers in stars to the flow of interstellar gas in stellar nurseries that span trillions of kilometers. Where mathematics failed, data-driven machine learning models provide a pathway to understanding turbulence and gaining insights into not only the origins and ultimate fate of stars and planetary ecosystems, but also opens pathways to nearly infinite sources of clean energy. Our StFT model presented in this paper is a stepping stone towards obviating the exponentially increasing computational costs of simulating geophysical systems (an inherently chaotic dynamical system requiring vast ensembles of high-resolution small time step simulations) as well as realization of digital twins to aid in the design and operational control of tokamak-based fusion power plants.

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

# A  Appendix

# B  Problem Setup and Datasets

## B.1  Plasma magnetohydrodynamic (MHD) equations

We consider magnetohydrodynamic (MHD) equations that characterize the plasma instabilities in fusion tokamaks. The coupled multi-physics system includes the continuity equation solving charge density $\delta n$, Poisson's equation solving $\delta\phi$, the Ampere's law to solving $\delta u_\parallel$, the Faraday's law with the assumption $E_\parallel = 0$ to solving $\delta A_\parallel$, and the perpendicular force balance equation to solving $\delta B_\parallel$. The first continuity equation

for gyrocenter charge density is expressed as,

$$
\begin{aligned}
&\frac{\partial \delta n}{\partial t} + \mathbf{B}_0 \cdot \nabla \left( \frac{n_0 \delta u_\parallel}{B_0} \right) - n_0 \mathbf{v}_* \cdot \frac{\nabla B_0}{B_0} + \delta \mathbf{B}_\perp \cdot \nabla \left( \frac{n_0 u_{\parallel 0}}{B_0} \right) \\
&- \frac{\nabla \times \mathbf{B_0}}{e B_0^2} \cdot \left( \nabla \delta P_\parallel + \frac{\left( \delta P_\perp - \delta P_\parallel \right) \nabla B_0}{B_0} \right) \\
&+ \nabla \cdot \left( \frac{\delta P_\parallel \mathbf{b}_0 \nabla \times \mathbf{b}_0 \cdot \mathbf{b}_0}{e B_0} \right) - \frac{\mathbf{b}_0 \times \nabla \delta B_\parallel}{e} \cdot \nabla \left( \frac{P_0}{B_0^2} \right) \\
&- \frac{\nabla \times \mathbf{b}_0 \cdot \nabla \delta B_\parallel}{e B_0^2} P_0 = 0,
\end{aligned}
\tag{3}
$$

where $n$ is the density, $B$ is the magnetic field, $u_\parallel$ is the parallel flow velocity, and $P$ is the pressure. The perturbed quantities are denoted by $\delta$ with the equilibrium states including temperature, density, magnetic field and the flux surface from the reconstruction of DIII-D experiments. Here, $\delta n = \delta n_e + q_i \delta n_i / q_e$ stands for the difference of ion and electron density, and $\delta u_\parallel = \delta u_{\parallel e} + q_i \delta u_{\parallel i} / q_e$ denotes the difference of ion and electron flow. We have $\mathbf{v}_* = \mathbf{b}_0 \times \nabla \left( \delta P_\parallel + \delta P_\perp \right) / (n_0 m_e \Omega_e)$, where $m_e$ is the electron mass, and $\Omega_e = e B_0 / m_e$ is the electron cyclotron frequency. The perturbed electron parallel flow $\delta u_\parallel$ can be solved from Ampere's law,

$$
\delta u_\parallel = \frac{1}{\mu_0 e n_0} \nabla_\perp^2 \delta A_\parallel,
\tag{4}
$$

where $\mu_0$ is the permeability of vacuum. $\delta A_\parallel$ is the perturbed vector potential. In the single fluid model, $E_\parallel = 0$ is assumed. Then $\delta A_\parallel$ can be solved from

$$
\frac{\partial A_\parallel}{\partial t} = \mathbf{b}_0 \cdot \nabla \phi,
\tag{5}
$$

and the electrostatic potential $\phi$ can be solved from gyrokinetic Poisson's equation (the quasi-neutrality condition)

$$
\frac{c^2}{v_A^2} \nabla_\perp^2 \phi = \frac{e \delta n}{\epsilon_0},
\tag{6}
$$

where $c$ is the speed of light, $v_A$ is the Alfvén velocity, and $\epsilon_0$ is the dielectric constant of vacuum. The parallel magnetic perturbation $\delta B$ is given by the perpendicular force balance,

$$
\frac{\delta B_\parallel}{B_0} = -\frac{\beta_e}{2} \frac{\delta P_\perp}{P_0} = -\frac{\beta_e}{2} \frac{\partial P_0}{\partial \psi_0} \frac{\delta \psi}{P_0}.
\tag{7}
$$

The perturbed pressure in the fluid limit can be calculated by

$$
\begin{aligned}
\delta P_\perp &= \frac{\partial P_0}{\partial \psi_0} \delta \psi - 2 \frac{\delta B_\parallel}{B_0} P_0, \\
\delta P_\parallel &= \frac{\partial P_0}{\partial \psi_0} \delta \psi - \frac{\delta B_\parallel}{B_0} P_0.
\end{aligned}
\tag{8}
$$

In these equations, $\psi_0$ and $\delta \psi$ is the equilibrium and perturbed magnetic flux, and the evolution of $\delta \psi$ is solved from

$$
\frac{\partial \delta \psi}{\partial t} = -\frac{\partial \phi}{\partial \alpha_0},
\tag{9}
$$

where $\alpha_0$ is from the Clebsch representation of $\mathbf{B}$ field, and $\mathbf{B}_0 = \nabla \psi_0 \times \nabla \alpha_0$. We run a linear gyrokinetic simulation with a $100 \times 250 \times 24$ mesh in radial, poloidal and parallel directions. The time step is set to $\Delta t = 0.005 R_0 / C_s = 1.483 \times 10^{-8}$ s. We keep both $n = 0, 1$ modes, generate a trajectory of $128,000$ time steps, and collect the data every 100 snapshots. We focus on emulating the dynamics of electrostatic potential $\delta \phi$, parallel vector potential $\delta A_\parallel$, electron number density $\delta n_e$, ion number density $\delta n_i$, and electron velocity $\delta u_e$ in their trajectories.

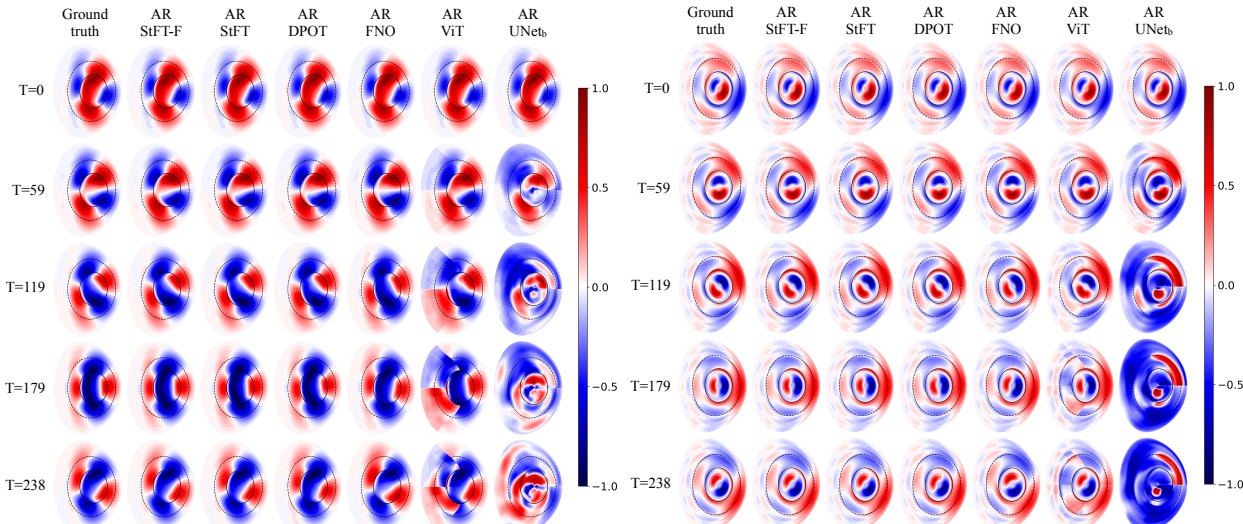

Figure 6: Temporal evolution of normalized perturbed parallel vector potential $\delta A_\parallel$ and perturbed electron density $\delta n_e$ contours predicted by different models: StFT-F, StFT, FNO, ViT and U-Net. Significant phase differences between the predictions of the models appear after $T = 59$, where StFT and StFT-F perform stable across the forecasting time horizon.

### B.2 2D incompressible Navier-Stokes equations

We consider the 2D incompressible Navier-Stokes (NS) equation on a rectangular domain $(x, y) \in [0, 1]^2$,

$$
\begin{aligned}
\frac{\partial u}{\partial t} + \frac{\partial p}{\partial x} &= -u\frac{\partial u}{\partial x} - v\frac{\partial u}{\partial y} + \frac{1}{\text{Re}}\nabla^2 u + f(x, y), \\
\frac{\partial v}{\partial t} + \frac{\partial p}{\partial y} &= -u\frac{\partial v}{\partial x} - v\frac{\partial v}{\partial y} + \frac{1}{\text{Re}}\nabla^2 v + f(x, y), \\
\frac{\partial u}{\partial x} + \frac{\partial v}{\partial y} &= 0,
\end{aligned}
\tag{10}
$$

where $u$ and $v$ represent the velocity components in the x and y directions, and $p$ represents the pressure. $f(x, y)$ is the source term, and we set it to $e^{-100((x-0.5)^2+(y-0.5)^2)}$. The Reynolds number is set to 1000. We run a finite difference solver to compute the solutions on a $50 \times 50$ spatial grid, with the temporal domain discretized into a total of 101 timestamps over $T \in [0, 20]$. We generated a total of 100 trajectories by sampling the four boundary conditions uniformly from $(0.1, 0.6)$.

### B.3 Spherical shallow-water equations

We consider the viscous shallow-water equations modeling the dynamics of large-scale atmospheric flows:

$$
\begin{aligned}
\frac{D\mathbf{V}}{Dt} &= -f\mathbf{k} \times \mathbf{V} - g\nabla h + \nu\nabla^2\mathbf{V}, \\
\frac{Dh}{Dt} &= -h\nabla \cdot \mathbf{V} + \nu\nabla^2 h, \quad x \in \Omega, \ t \in [0, 1],
\end{aligned}
\tag{11}
$$

where $\mathbf{V}$ is the velocity vector tangential to the spherical surface, $\mathbf{k}$ is the unit vector normal to the surface, $h$ is the thickness of the fluid layer, $f = 2\Xi\sin\phi$ is the Coriolis parameter ($\Xi$ being the Earth's angular velocity), $g$ is the gravitational acceleration, and $\nu$ is the diffusion coefficient. The equations are defined over a spherical domain $\Omega = (\lambda, \phi)$, with longitude $\lambda$ and latitude $\phi$.

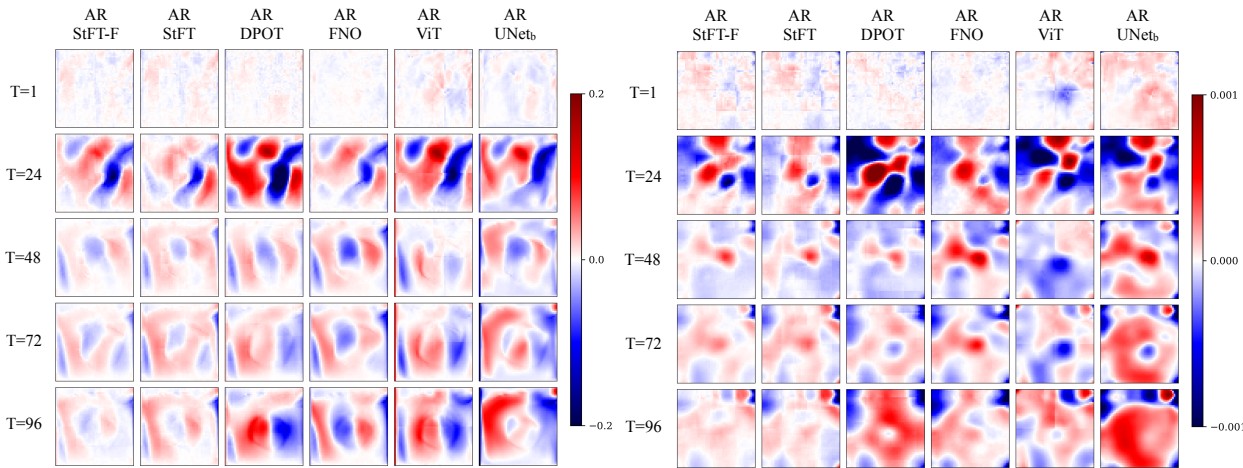

Figure 7: 2D incompressible Navier-Stokes equation: pointwise error of the predicted evolution of velocity component $u$ and pressure $p$ contours across different models: StFT-F, StFT, FNO, ViT and U-Net. For long-term predictions, StFT and StFT-F demonstrate lower residuals compared to other models.

As an initial condition, a zonal flow typical of a mid-latitude tropospheric jet is defined for the velocity component $u$ as a function of latitude $\phi$:

$$
u(\phi, t=0) = \begin{cases} 0, & \phi \leq \phi_0, \\ \frac{u_{\max}}{n} \exp\left[\frac{1}{(\phi-\phi_0)(\phi-\phi_1)}\right], & \phi_0 < \phi < \phi_1, \\ 0, & \phi \geq \phi_1, \end{cases}
$$

where $u_{\max}$ is the maximum zonal velocity, $\phi_0$ and $\phi_1$ represent the southern and northern boundaries of the jet in radians, and $n = \exp[-4/(\phi_1 - \phi_0)^2]$ normalizes $u_{\max}$ at the midpoint of the jet. To induce barotropic instability, a localized Gaussian perturbation is added to the height field, expressed as:

$$
h'(\lambda, \phi, t=0) = \hat{h} \cos(\phi) \exp\left[-\left(\frac{\lambda}{\alpha}\right)^2\right] \exp\left[-\left(\frac{\phi_2 - \phi}{\beta}\right)^2\right],
$$

where $-\pi < \lambda < \pi$, and parameters $\hat{h}, \phi_2, \alpha$, and $\beta$ control the shape and location of the perturbation. The parameters $\alpha$ and $\beta$ are sampled from uniform distributions $\alpha \sim U[0.1, 0.5]$ and $\beta \sim U[0.03, 0.2]$. We ran the solver from Dedalus Burns et al. (2020) on a $256 \times 256$ spherical grid, and the temporal dimension is discretized into 72 timestamps. We have a total of 200 trajectories by sampling $\alpha$ and $\beta$.

## C  Ablation Study

### C.1  The Hierarchical Structure and The Frequency Path

To evaluate the effectiveness of the hierarchical structure and the frequency path in StFT, we conduct an ablation study on the analysis of the joint influence and trade-off between patch size and the number of retained Fourier modes. First, we only keep one layer of StFT while changing the number of Fourier modes $k$. Second, we keep the hierarchical structure, and tuning $k$ in each hierarchical layer.

Table 4 shows the $L_2$ relative errors averaged over all the variables. Note that $l_1$ is the coarsest level, and $l_3$ is the finest level. $\mathcal{F}$ stands for the frequency path in StFT blocks. These results demonstrate the effectiveness of the hierarchical composition of StFT blocks and the frequency component in StFT block. With the hierarchical composition, for Plasma MHD, the error drops to 0.0307 from 0.0805, and for shallow-water equations (SWE), the error drops to 0.0625 from 0.0975 with $k = 8$ retained modes in the Fourier space. We observe that the

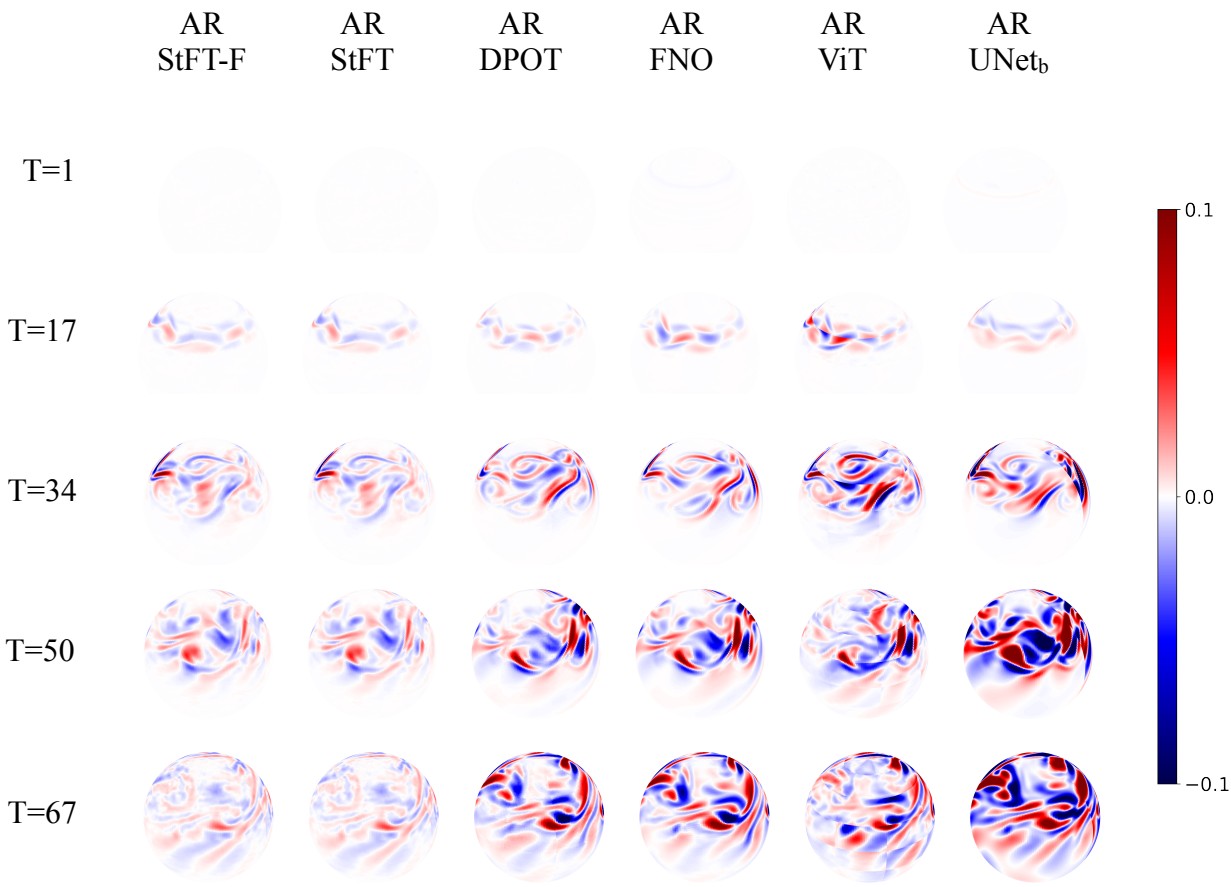

Figure 8: Spherical shallow-water equations: pointwise error of the temporal evolution of velocity field predicted by all the autoregressive models: StFT-F, StFT, FNO, ViT and U-Net. The prediction error exhibits a temporal growth trend, with our model StFT and StFT-F consistently demonstrate lower residuals over the forecasting time horizon.

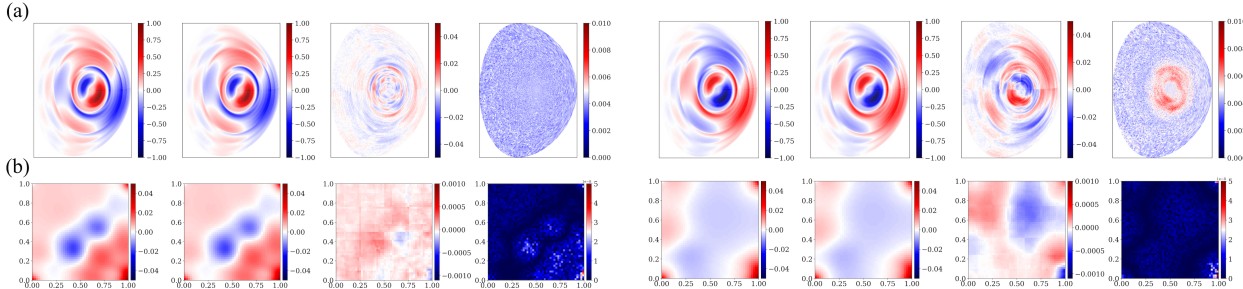

Figure 9: Additional evaluation of forecasting: ground truth, StFT-F prediction, residual, and uncertainty over time - shown for initial (left) and final (right) states. Variables include: (a) $\delta n_e$ in plasma MHD, (b) $p$ in Navier-Stokes.

| Model | (Multi-) Scale | Fourier Modes in $\mathcal{F}$ | | |
| Dataset | Patch Size | $k=4$ | $k=8$ | $k=16$ |
|---|---|---|---|---|
| MHD | $l_1$: (128,128) | 0.0926 | 0.0805 | 0.0542 |
| MHD | $l_2$: (64,64) | 0.1206 | 0.1050 | 0.1131 |
| MHD | $l_1 + l_2$: ((128,128),(64,64)) | 0.0544 | **0.0307** | 0.0578 |
| SWE | $l_1$: (128,128) | 0.5110 | 2.5729 | 47.178 |
| SWE | $l_2$: (64,64) | 0.1123 | 0.1010 | 0.14744 |
| SWE | $l_3$: (32,32) | 0.0984 | 0.0975 | 0.5455 |
| SWE | $l_1 + l_2 + l_3$: ((128,128),(64,64),(32,32)) | 0.0766 | **0.0625** | 0.0829 |

Table 4: Ablation study results. We run models with combinations of StFT blocks and the retained Fourier modes in the frequency path. $l_1$ is the coarsest level, and $l_3$ is the finest level. $k$ stands for the number of retained Fourier modes in the frequency path of StFT blocks.

fine-level layer setting in the SWE achieves the best performance among the single layer results, and the multi-layer settings further decrease the prediction error. Higher Fourier modes contribute substantially to the coarse-layer representations in the MHD system, reflecting perturbations that are distributed over a broader spatial domain. In contrast, for the SWE problem, high frequency components are more spatially localized, predominantly confined to small regions. The interaction between refinement level and Fourier modes indicate that the model performance generally improves with the hierarchical composition, while the retained Fourier modes must be tuned to match the dominant frequency content present within each StFT block.

## C.2 Convergence on the Number of Scales

We evaluate the performance of StFT on the number of scales for the plasma MHD dataset. Specifically, we run StFT with one scale (patch size of 128), three scales (patch sizes of 128, 64, and 48), and four scales (patch sizes of 128, 64, 48, and 32). As shown in Table 7, it is observed that increasing the number of scales leads to a reduction in prediction error to a certain point, with the two-layer configuration achieving the lowest error (0.0307). Beyond this, the inclusion of additional scales results in a slight increase in error, indicating a convergence in model performance, and further increasing the number of scales may not provide performance gain.

Table 5: Effect of the number of scales on L2 relative error of the overall prediction on the plasma MHD.

| Number of Scales | L2 Relative Error |
|---|---|
| 1 | 0.0805 |
| 2 | 0.0307 |
| 3 | 0.0385 |
| 4 | 0.0391 |

## C.3 The Overlapping Tokenizer

We conduct an ablation study on the overlapping tokenizer design, evaluating both predictive performance and computational cost, as reported in Tables 5 and 6. We evaluate the computational cost including the inference FLOPS per sample, training/inference time per batch and peak memory usage. We used a fixed batch size of 20 for all models. The results are summarized in the following two tables. StFT-O stands for StFT with the overlapping tokenizer and StFT-NO for the opposite. All experiments were performed on an NVIDIA A100 GPU. Incorporating the overlapping tokenizer leads to substantial improvements in accuracy, in plasma MHD and shallow-water datasets, with errors reductions of 68% and 10% respectively.

Notably, this enhancement comes with negligible impact on computational complexity, inference time and peak memory usage.

Table 6: Effect of the overlapping tokenizer on the prediction error of plasma MHD and shallow-water datasets.

| Method | Plasma MHD | Shallow-Water |
|---|---|---|
| With overlapping tokenizer | **0.0307** | **0.0625** |
| Without overlapping tokenizer | 0.0986 | 0.0700 |

Table 7: Computational complexity comparison of StFT with and without overlapping tokenizer.

| Dataset | Method | GFLOPs | Training / Iter (s) | Inference / Iter (s) | Peak Memory (GB) |
|---|---|---|---|---|---|
| MHD | StFT-O | 0.704 | 0.181 | 0.0455 | 9.49 |
| MHD | StFT-NO | 0.704 | 0.157 | 0.0454 | 9.49 |
| SWE | StFT-O | 4.30 | 0.189 | 0.0319 | 12.2 |
| SWE | StFT-NO | 4.30 | 0.189 | 0.0319 | 12.2 |

## D  Computational and Space Complexity

We compare StFT with other baselines regarding the inference FLOPS per sample, training time per batch, inference time per batch, and peak memory usage (during training). In plasma MHD and shallow-water datasets, compared to FNO, StFT is 20x smaller, and 8x smaller in computation (FLOPS) respectively. While achieving the highest prediction accuracy, StFT has the same order of magnitude of FLOPs as UNet and ViT in plasma MHD. In the shallow-water equation (SWE), however UNet is much more computationally expensive - 11x more than StFT – while its relative error is 3x larger than StFT. StFT is also efficient in both training and inference time. For plasma MHD, StFT achieves a 30% reduction in training time and 50% reduction in inference time compared to FNO. For SWE, StFT is 40% faster in training and 60% faster than FNO in inference. Although StFT incorporates dual paths operating in the frequency domain and the spatio-temporal domain, its peak memory usage remains comparable to that of FNO in the plasma MHD, and is reduced by 56% compared to FNO in SWE, and 26% less than UNet in SWE. Figure 10 compares the inference time and FLOPs versus mean L2 relative error for all models.

Table 8: Comparison of methods on plasma MHD and shallow-water datasets regarding the computational and space complexity.

| Dataset | Method | GFLOPs | Training / Iter (s) | Inference / Iter (s) | Peak Memory (GB) |
|---|---|---|---|---|---|
| MHD | StFT | 0.704 | 0.181 | 0.0455 | 9.49 |
| MHD | DPOT | 1.10 | 0.0676 | 0.0120 | 3.20 |
| MHD | FNO | 13.8 | 0.262 | 0.0900 | 10.1 |
| MHD | UNET | 0.462 | 0.0263 | 0.00230 | 1.38 |
| MHD | ViT | 0.338 | 0.0171 | 0.00237 | 1.65 |
| SWE | StFT | 4.30 | 0.189 | 0.0319 | 12.2 |
| SWE | DPOT | 1.61 | 0.0427 | 0.0133 | 8.43 |
| SWE | FNO | 34.6 | 0.325 | 0.0886 | 30.2 |
| SWE | UNET | 48.5 | 0.107 | 0.0257 | 16.5 |
| SWE | ViT | 0.90 | 0.0130 | 0.00347 | 7.45 |

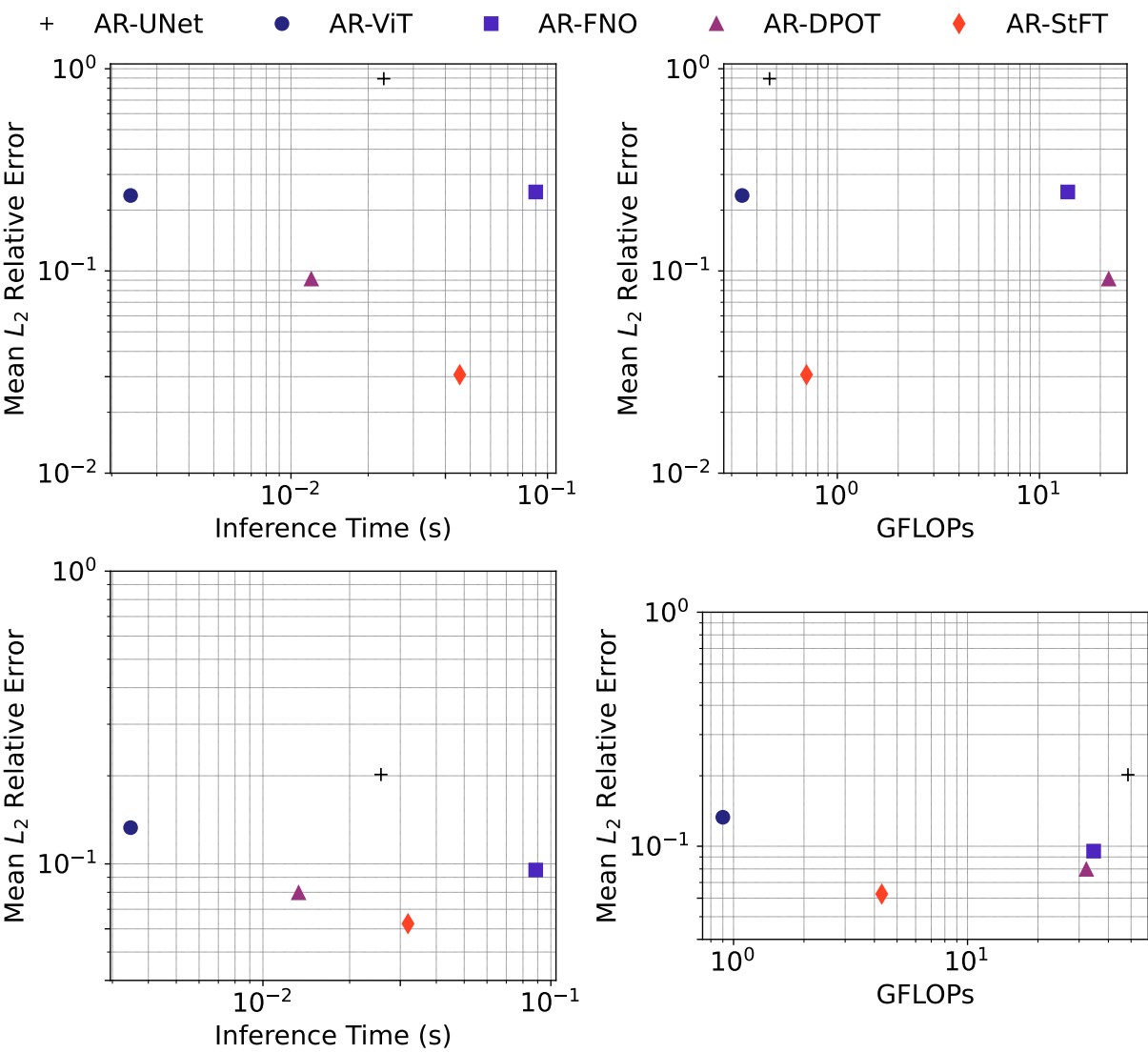

Figure 10: Comparison of models in terms of mean L2 relative error versus inference time (left) and FLOPs (right) for plasma MHD (top) and shallow-water (bottom).

# E  Signficance of UQ on StFT-F

For the Plasma MHD dataset, we measure the average coverage for the first 20 autoregressively predicted snapshots (each row represents one physical variable), as shown in Table 9. In the last row, we report the average coverage across all physical variables. For the 90% confidence interval, average coverage is very close to the ideal value of 90%, demonstrating that StFT-F is well-calibrated around the 90% confidence interval. For the 95% confidence interval, the average coverage is under-confident by 2.5%, suggesting the intervals may be slightly narrow. For the shallow-water dataset, we provide the results in Table 10, where we measure the average coverage for the first 10 predicted snapshots. For the 90% confidence interval, average coverage is also very close to the ideal value of 90%, which shows that StFT-F is also well-calibrated for the shallow-water dataset.

Table 9: Confidence interval coverage on the plasma MHD dataset.

| Variable | CI: 90% | CI: 95% |
|---|---|---|
| $\delta\phi$ | 0.906 | 0.937 |
| $\delta A_{\parallel}$ | 0.977 | 0.988 |
| $\delta B_{\parallel}$ | 0.900 | 0.930 |
| $\delta n_e$ | 0.933 | 0.962 |
| $\delta n_i$ | 0.867 | 0.910 |
| $\delta u_e$ | 0.781 | 0.823 |
| **Average Coverage** | **89.4%** | **92.5%** |

Table 10: Confidence interval coverage on the shallow-water dataset.

| Variable | CI: 90% | CI: 95% |
|---|---|---|
| **V** | 0.895 | 0.980 |

Table 11: Training/validation/test data splits and the training budget for all models measured on an A100 GPU.

| Dataset | Total | Split (Train / Val / Test) | Training Budget |
|---|---|---|---|
| Plasma MHD | 1 traj. (1,221 snapshots) | 927 / 50 / 244 | 24h |
| Navier-Stokes | 100 traj. (101 snaps/traj) | 80 / 10 / 10 | 24h |
| Shallow-water | 200 traj. (72 snaps/traj) | 160 / 20 / 20 | 48h |

# F  Experimental Details

**Training/validation/test data sets.**  For the plasma MHD data, we split the trajectory of 1221 snapshots into a training set (the first 927 snapshots), a validation set (the middle 50 snapshots), and a test set (the last 244 snapshots). For the Navier-Stokes dataset, we have a total of 100 trajectories (101 snapshots for each trajectory), and split them into 80 trajectories for training, 10 for validation, and 10 for testing. For the shallow-water dataset, we have a total of 200 trajectories (72 snapshots for each trajectory), and split them into 160 trajectories for training, 20 for validation, and 20 for testing. To ensure a fair comparison, we impose a fixed training budget across all models. Specifically, a 48-hour limit mesaured on one A100 GPU was set for the shallow-water equation dataset, while a 24-hour limit is applied to both the plasma MHD and Navier-Stokes equation datasets. Table 11 summarizes the datasets and the training budget for all models.

**Generative residual correction block.**  We follow a two-step training protocol in training StFT-F: first, we train StFT thoroughly with the training budget, and then we train the generative residual correction block

for another 200 epochs. We employ a rectified flow to learn distributions of the residuals given the prediction of StFT and the history snapshots. We implement a similar structure to the Diffusion Transformer (DiT) as the backbone model (Peebles & Xie, 2023). In each DiT block, we apply adaptive layer normalization before a self-attention layer and an MLP layer. We use adaLN-Zero for time conditioning. For the history snapshots $\tilde{u}_t$ and the prediction $\mathcal{F}_{\theta_d}(\tilde{u}_t)$ of StFT, these conditions are incorporated as extra input tokens.

**Hyperparameters.** For StFT on the plasma dynamics, 3D FFT is used to encode the spatio-temporal inputs in the frequency path. We use the patch size of 128 for the the first StFT block and and 64 for the second StFT block. The overlapping size is set to 1. The hidden dimension is set to 128. The depth for each StFT block is set to 6. We keep the lowest 8 frequencies for each spatial dimension. For the rectified flow block, the depth is set to 8, and the hidden dimension is set to 128. For the Navier-Stokes equation, StFT uses a patch size of 25 for the coarse block (the first block), 13 for the middle block, and 8 for the last block. The overlapping size is set to 0, and the frequency path is not used. For each block in the hierarchical structure, the depth is set to 8, and we use a hidden dimension of 512. In the rectified flow block, we use a depth of 4 and set the hidden dimension to 128. For the shallow-water equation, three levels of StFT blocks are employed, and their patch sizes are set to 128, 64, and 32 respectively. For each block, the depth is set to 6, and the hidden dimension is set to 512. We use the 2D FFT to encode the spatio-temporal inputs, and the lowest 8 frequencies are kept for each spatial dimension. The overlapping size is set to 1. For the rectified flow model, we use a depth of 8 and a hidden dimension of 128.

Table 12: Hyperparameter search range for each method.

| Method | Hyperparameter Search |
|---|---|
| **DPOT** | Hidden dimension: $[256, 512]$ 
 Patch size: $[8, 16, 32]$ 
 Depth: 6 
 Heads: 4 |
| **AR-FNO** | Modes: $[16, 20, 24]$ 
 Layers: $[4, 5]$ 
 Hidden dimension: 256 |
| **AR-ViT** | Hidden dimension: $[256, 512]$ 
 Patch size: $[16, 32, 64]$ |
| **AR-UNet** | Bottleneck hidden dimension: $[64, 128, 256, 512]$ |

**Baselines.** For all the baselines, we run all models with the same training time budget, as detailed in Table 11. For DPOT, we vary the hidden dimension from $[256, 512]$, and the patch size from $[8, 16, 32]$. We set the depth as 6, and the number of heads as 4, which is the default setting from the author's implementation. For AR-FNO, the number of modes are selected through a search in $[16, 20, 24]$, the number of layers are searched in $[4, 5]$, and the hidden dimension is set to 256. For AR-ViT, we vary the hidden dimension from $[256, 512]$, and the patch size from $[16, 32, 64]$. For AR-UNet, the hidden dimension of bottleneck embeddings are searched in $[64, 128, 256, 512]$. Table 12 summarizes the hyperparameter choices for each method.

## G More Visualization Results

Figure 6 illustrates the ground truth and predicted temporal evolution of normalized perturbed parallel vector potential $\delta A_\parallel$ and perturbed electron density $\delta n_e$ in plasam MHD using AR-StFT, AR-StFT-F, AR-DPOT, AR-FNO, AR-ViT and AR-UNet methods. StFT and StFT-F perform stable across the forecasting time horizon. Figure 7 and 8 show the pointwise error of all the models compared to the ground truth data in the Navier-Stokes and shallow-water equations, where StFT and StFT-F demonstrate lower residuals compared to other baseline models.

In addition, we compare StFT with StFT-F for the autoregressive prediction across different timestamps for velocity $u$ in Navier-Stokes equation, shown in Figure 11. A slightly larger error is observed between $t = 0.4$ and $0.6$, which can be attributed to the training objective of StFT. As StFT is optimized using a pointwise loss function, it is encouraged to produce predictions that closely match the most likely outcome.

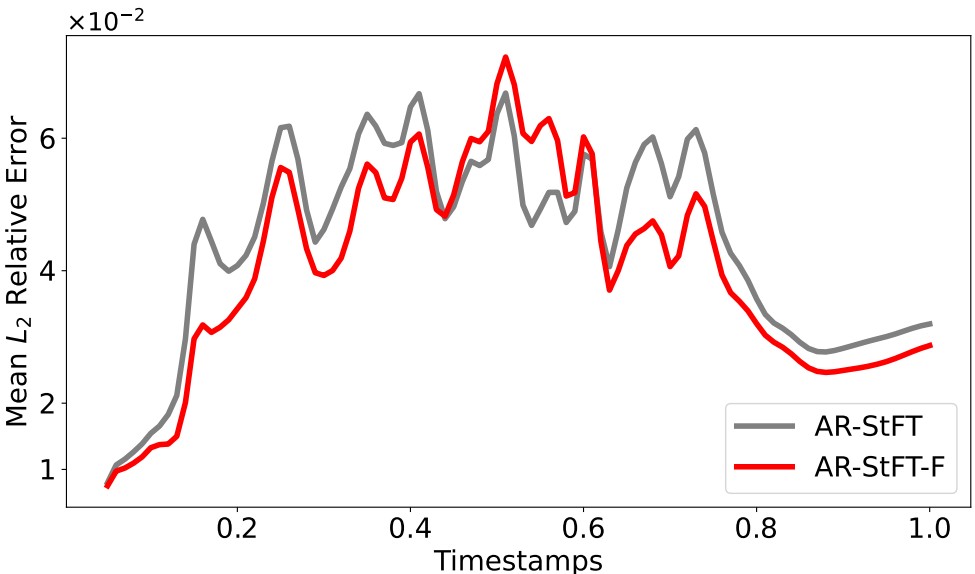

Figure 11: Results of comparing StFT and StFT-F for the autoregressive prediction in $L_2$ relative error across different timestamps for velocity $u$ in Navier-Stokes equation. The shaded region represents the standard deviation distribution of the relative error of StFT-F. However, the uncertainty is negligible that it is not visually discernible. StFT-F demonstrates better performance in the latter stages of the forecasting time horizon compared to StFT.

In contrast, StFT-F is designed to learn the full distribution of the target, potentially introducing higher variance in its predictions. This distributional modeling, while beneficial for uncertainty quantification, may result in marginally increased errors as it captures characteristics beyond the mean behavior.

