# OpenReview forum: "StFT: Spatio-temporal Fourier Transformer for Long-term Dynamics Prediction"
_TMLR — Accepted by TMLR_

### Review · Reviewer_yNL3 · 2025-10-14

**Summary Of Contributions:**

The paper introduces StFT (Spatio-temporal Fourier Transformer) and its probabilistic extension StFT-F for forecasting the evolution of spatial fields and uncertainty quantification. The model block combines a frequency path to capture low-frequency structure and a spatio-temporal path for local details. Predictions are produced from the past k snapshots of the variables and refined hierarchically from coarse to fine scales. Additionally, a rectified flow-matching block is trained on the residuals between the deterministic prediction and the true next state, enabling uncertainty quantification. Experiments on three spatio-temporal multi-physics systems compare StFT-F with autoregressive baselines such as AR-FNO, AR-UNet, and AR-ViT, AR-DPOT.

Strengths
-  The multiscale hierarchy can help reduce compounding error.
-  The model leverages spectral and spatial representations.
-  The rectified residual block provides additional refinement and uncertainty estimation.
-  Numerical results show improved performance compared to benchmarks.
-  The paper is easy to follow overall.

Weeknesses
-  Baselines do not include recent models (e.g., PDE-Refiner or Dyffusion) despite being mentioned in the introduction.
-  Multiple overlapping components (spectral, temporal, diffusion) make it difficult to isolate the contribution of each.
-  The interplay between spatial resolution in the hierarchical refinement and the number of retained Fourier components is not clear.
-  Uncertainty quantification is performed by sampling from the residual network, which can yield a different mean prediction than propagation of the deterministic mean.
- Results on uncertainty calibration are limited, no comparison with recent models with UQ.

**Audience:**

No

**Audience Explanation:**

While the empirical results are promising, the study omits comparisons with established methods. The work requires substantial revision and broader evaluation.

**Broader Impact Concerns:**

no concerns

**Claims And Evidence:**

No

**Claims Explanation:**

- Limited baseline coverage.  The baselines do not include recent models with uncertainty quantification (e.g., PDE-Refiner, DeepONet-Grid-UQ), even though such approaches are discussed in the introduction. As a result, the comparison focuses on standard deterministic autoregressive models and may not fully reflect the current state of the field.
- Unclear contribution of individual components.  The model combines spectral, spatial–temporal, and diffusion modules, but their respective roles and interactions are not clear.
- Interplay between spatial resolution and Fourier truncation.  The relationship between the spatial resolution used in the hierarchical refinement and the number of retained Fourier components is not analyzed. Since both affect the model’s effective bandwidth, their joint influence on stability and accuracy is unclear.
- Uncertainty estimation based on a separate rollout regime.  Uncertainty quantification is performed by sampling from the residual network, which can yield a different mean trajectory from the mean propagation.
- Limited analysis of uncertainty calibration and temporal dependence.  It is only quantified in terms of coverage %, e.g. no ECE is given.  Also it is not clear (qualitatively) how it propagates in time.

**Requested Changes:**

- Include proper comparison with methods (e.g., PDE-Refiner, Dyffusion, see above).
- Explain the relation between patch size, refinement level, and number of retained Fourier modes.
- Specify that the flow-matching residual refines one-step predictions; UQ sampling may shift the mean.
- Add ablations isolating spatial, spectral, and residual components.
- Provide quantitative UQ and calibration analysis over time and comparison with other methods.

---

### Review · Reviewer_Cb2H · 2025-10-15

**Summary Of Contributions:**

This paper presents StFT (Spatio-temporal Fourier Transformer), a hierarchical transformer framework for long-term forecasting of multi-scale physical dynamics governed by PDEs. Each StFT block models global dependencies through a Fourier-domain path and local dynamics through a spatio-temporal path, capturing interactions across spatial and temporal scales. The method also introduces an overlapping tokenizer to ensure smooth patch transitions and a probabilistic extension, StFT-F, that employs flow matching for residual correction and calibrated uncertainty quantification.

Extensive experiments on diverse PDE systems (magnetohydrodynamics, Navier–Stokes, and shallow-water equations) demonstrate substantial improvements in long-horizon accuracy and stability over strong Fourier and transformer baselines.

**Audience:**

Yes

**Audience Explanation:**

The paper would interest TMLR readers working on scientific machine learning, spatio-temporal modeling, and physics-informed deep learning, as it advances long-term dynamics prediction and uncertainty quantification.

**Claims And Evidence:**

Yes

**Claims Explanation:**

The claims are well supported by clear and comprehensive evidence. Experiments on three PDE-based benchmarks show consistent gains in long-term accuracy, stability, and uncertainty calibration over strong baselines. Ablation studies validate the role of key components, and results are clearly presented. Minor details on computational cost could be expanded, but overall, the evidence convincingly supports the paper’s claims.

**Requested Changes:**

While the paper demonstrates strong predictive performance, it provides limited quantitative discussion of computational cost. Expanding this section would make the contribution more transparent and practically relevant. The authors should report runtime, memory usage, and parameter counts compared to key baselines (e.g., FNO, AFNO, and transformer variants) under identical settings. A scaling analysis showing how training and inference time grow with grid resolution, sequence length, and the number of StFT layers would also help readers assess the method’s efficiency for large-scale or real-time simulations. Including these comparisons, either in the main text or supplementary material, would strengthen the empirical rigor of the paper.

---

### Review · Reviewer_js98 · 2025-10-26

**Summary Of Contributions:**

The paper introduces StFT, a spatio-temporal Fourier Transformer for long-term forecasting of multi-scale, multi-physics PDE systems. Each StFT block models dynamics at a distinct spatial scale using a dual-path design—one path operates in the Fourier domain and the other in the spatio-temporal domain. The blocks are composed coarse-to-fine to explicitly encode cross-scale interactions. Evaluations on plasma MHD, 2D incompressible Navier–Stokes, and spherical shallow-water show improved long-horizon accuracy.

Strengths

1. Clear, modular architecture that separates global (frequency) and local (spatio-temporal) processing; hierarchy aligns well with PDE multi-scale structure.
2. Practical overlapping tokenizer to improve spatial continuity with negligible overhead (as shown in ablations).
3. Uncertainty quantification via rectified flow yields useful, empirically calibrated intervals on multiple datasets.

Weaknesses

1. Plasma experiments appear to rely on a single long trajectory, limiting cross-condition generalization evidence (Table 11).
2. The proposed method currently targets regular grids; authors note this as a limitation.
3. Some tasks show slightly higher point error at certain horizons due to distributional modeling.

**Audience:**

Yes

**Audience Explanation:**

TMLR’s readership includes researchers in scientific ML, operator learning, forecasting, and uncertainty quantification. The paper advances multi-scale neural operators with an interpretable hierarchy and introduces a pragmatic UQ mechanism for long-horizon rollouts, both of broad interest to this community.

**Broader Impact Concerns:**

No concerns.

**Claims And Evidence:**

Yes

**Claims Explanation:**

The core architectural claims are well-substantiated: the paper explains the dual-path block, hierarchical composition, and overlapping tokenizer clearly, with algorithms and diagrams. Empirically, the autoregressive setup is consistent across methods; budgets and baselines are described, and long rollouts are used.

**Requested Changes:**

1. The plasma evaluation is based on a single trajectory split into train/val/test. Please add experiments across multiple discharges/conditions if possible.
2. Expand ablations to cover: (i) frequency truncation bandwidth, (ii) patch sizes/overlap hyperparameters, and (iii) number of scales beyond the brief convergence table.
3. Consider adding (a) a non-autoregressive baseline (e.g., continuous-time emulator trained for multi-step), and (b) a physics-informed operator (if feasible). Current baselines are strong but largely data-driven.

---

### Decision · Action_Editor_vWPE · 2025-12-22

**Recommendation:** Accept with minor revision

**Additional Comments:**

According to Reviewer yNL3, the requested ablation studying the relationship between patch size, refinement level, and the number of retained Fourier modes is still missing. In the final revision, please provide an analysis of the joint influence and trade-off between patch size and the number of retained Fourier modes on model performance.

**Audience:**

Yes

**Audience Explanation:**

Certainly, there is considerable audience working in spatio-temporal forecasting using machine learning interested in the paper's findings.

**Claims And Evidence:**

Yes

**Claims Explanation:**

The paper introduces a novel method called Spatio-temporal Fourier Transformer for long-term dynamics prediction, as the authors note that the method is particularly suited for long-term forecasting of systems characterized by multi-scale behaviors involving dynamics of different orders. The method is thoroughly evaluated against several state-of-the-art methods using a diverse set of simulated datasets (derived from plasma, fluid, and atmospheric dynamics). Some predictions are visualized, providing a quantitative comparison of the predictions against the ground truth. Ablations are provided to further the understanding of the method's foundations.

All of the reviewers are in agreement to accept the paper, however, some minor edits are requested by one of the reviewers (see below).
Final Decision: Accept with minor revision.